# Premise Selection for Theorem Proving
# by Deep Graph Embedding

**Mingzhe Wang**[*]   **Yihe Tang**[*]   **Jian Wang**   **Jia Deng**
University of Michigan, Ann Arbor

## Abstract

We propose a deep learning-based approach to the problem of premise selection: selecting mathematical statements relevant for proving a given conjecture. We represent a higher-order logic formula as a graph that is invariant to variable renaming but still fully preserves syntactic and semantic information. We then embed the graph into a vector via a novel embedding method that preserves the information of edge ordering. Our approach achieves state-of-the-art results on the HolStep dataset, improving the classification accuracy from 83% to 90.3%.

## 1   Introduction

Automated reasoning over mathematical proofs is a core question of artificial intelligence that dates back to the early days of computer science [1]. It not only constitutes a key aspect of general intelligence, but also underpins a broad set of applications ranging from circuit design to compilers, where it is critical to verify the correctness of a computer system [2, 3, 4].

A key challenge of theorem proving is *premise selection* [5]: selecting relevant statements that are useful for proving a given conjecture. Theorem proving is essentially a search problem with the goal of finding a sequence of deductions leading from presumed facts to the given conjecture. The space of this search is combinatorial—with today's large mathematical knowledge bases [6, 7], the search can quickly explode beyond the capability of modern automated theorem provers, despite the fact that often only a small fraction of facts in the knowledge base are relevant for proving a given conjecture. Premise selection thus plays a critical role in narrowing down the search space and making it tractable.

Premise selection has been mainly tackled as hand-designed heuristics based on comparing and analyzing symbols [8]. Recently, some machine learning methods have emerged as a promising alternative for premise selection, which can naturally be cast as a classification or ranking problem. Alama et al. [9] trained a kernel-based classifier using essentially bag-of-words features, and demonstrated large improvement over the state of the art system. Alemi et al. [5] were the first to apply deep learning approaches to premise selection and demonstrated competitive results without manual feature engineering. Kaliszyk et al. [10] introduced HolStep, a large dataset of higher-order logic proofs, and provided baselines based on logistic regression and deep networks.

In this paper we propose a new deep learning approach to premise selection. The key idea of our approach is to represent mathematical formulas as graphs and embed them into vector space. This is different from prior work on premise selection that directly applies deep networks to sequences of characters or tokens [5, 10]. Our approach is motivated by the observation that a mathematical formula can be represented as a graph that encodes the syntactic and semantic structure of the formula. For example, the formula $\forall x \exists y (P(x) \land Q(x, y))$ can be expressed as the graph shown in Fig. 1, where edges link terms to their constituents and connect quantifiers to their variables.

---

[*]Equal contribution.

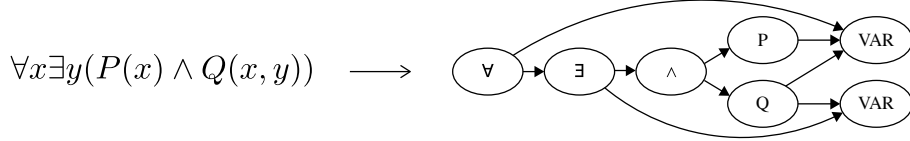

Figure 1: The formula $\forall x \exists y (P(x) \wedge Q(x,y))$ can be represented as a graph.

Our hypothesis is that such graph representations are better than sequential forms because a graph makes explicit key syntactic and semantic structures such as composition, variable binding, and co-reference. Such an explicit representation helps the learning of invariant feature representations. For example, $P(x, T(f(z) + g(z), v)) \wedge Q(y)$ and $P(y) \wedge Q(x)$ share the same top level structure $P \wedge Q$, but such similarity would be less apparent and harder to detect from a sequence of tokens because syntactically close terms can be far apart in the sequence.

Another benefit of a graph representation is that we can make it invariant to variable renaming while preserving the semantics. For example, the graph for $\forall x \exists y (P(x) \wedge Q(x,y))$ (Fig. 1) is the same regardless of how the variables are named in the formula, but the semantics of quantifiers and co-reference is completely preserved—the quantifier $\forall$ binds a variable that is the first argument of both $P$ and $Q$, and the quantifier $\exists$ binds a variable that is the second argument of $Q$.

It is worth noting that although a sequential form encodes the same information, and a neural network may well be able to learn to convert a sequence of tokens into a graph, such a neural conversion is unnecessary—unlike parsing natural language sentences, constructing a graph out of a formula is straightforward and unambiguous. Thus there is no obvious benefit to be gained through an end-to-end approach that starts from the textual representation of formulas.

To perform premise selection, we convert a formula into a graph, embed the graph into a vector, and then classify the relevance of the formula. To embed a graph into a vector, we assign an initial embedding vector for each node of the graph, and then iteratively update the embedding of each node using the embeddings of its neighbors. We then pool the embeddings of all nodes to form the embedding of the entire graph. The parameters of each update are learned end to end through backpropagation. In other words, we learn a deep network that embeds a graph into a vector; the topology of the unrolled network is determined by the input graph.

We perform experiments using the HolStep dataset [10], which consists of over two million conjecture-statement pairs that can be used to evaluate premise selection. The results show that our graph-embedding approach achieves large improvement over sequence-based models. In particular, our approach improves the state-of-the-art accuracy on HolStep by 7.3%.

Our main contributions of this work are twofold. First, we propose a novel approach to premise selection that represents formulas as graphs and embeds them into vectors. To the best our knowledge, this is the first time premise selection is approached using deep graph embedding. Second, we improve the state-of-the-art classification accuracy on the HolStep dataset from 83% to 90.3%.

## 2 Related Work

Research on automated theorem proving has a long history [11]. Decades of research has resulted in a variety of well-developed automated theorem provers such as Vampire [12] and E [13]. However, no existing automated provers can scale to large mathematical libraries due to combinatorial explosion of the search space. This limitation gave rise to the development of interactive theorem proving [11] such as Coq [14] and Isabelle [15], which combines humans and machines in theorem proving and has led to impressive achievements such as the proof of the Kepler conjecture [16] and the formal proof of the Feit-Thompson problem [17].

Premise selection as a machine learning problem was introduced by Alama et al. [9], who constructed a corpus of proofs to train a kernelized classifier using bag-of-word features that represent the occurrences of terms in a vocabulary. Deep learning techniques were first applied to premise selection in the DeepMath work by Alemi et al. [5], who applied recurrent networks and convolutional to formulas represented as textual sequences, and showed that deep learning approaches can achieve competitive results against baselines using hand-engineered features. Serving the needs for large

datasets for training deep models, Kaliszyk et al. [10] introduced the HolStep dataset that consists of 2M statements and 10K conjectures, an order of magnitude larger than the DeepMath dataset [5].

A related task to premise selection is *internal guidance* of ATPs [18, 19, 20, 21, 22, 23, 24], the selection of the next clause to process *inside* an automated theorem prover. Internal guidance differs from premise selection in that internal guidance depends on the logical representation, inference algorithm, and current state inside a theorem prover, whereas premise selection is only about picking relevant statements as the initial input to a theorem prover that is treated as a black box. Because internal guidance is tightly integrated with proof search and is invoked repeatedly, efficiency is as important as accuracy, whereas for premise selection efficiency is not as critical.

Loos et al. [25] were the first to apply deep networks to internal guidance of ATPs. They experimented with both sequential representations and tree representations (recursive neural networks [26, 27]). Note that their tree representations are simply the parse trees, which, unlike our graphs, are not invariant to variable renaming and do not capture how quantifiers bind variables. Whalen [23] uses GRU networks to guide the exploration of partial proof trees, with formulas represented as sequences of tokens.

In addition to premise selection and internal guidance, other aspects of theorem proving have also benefited from machine learning. For example, Kühlwein & Urban [28] applied kernel methods to strategy finding, the problem of searching for good parameter configurations for an automated prover. Similarly, Bridge et al. [29] applied SVM and Gaussian Processes to select good heuristics, which are collections of standard settings for parameters and other decisions.

Our graph embedding method is related to a large body of prior work on embeddings and graphs. Deepwalk [30], LINE [31] and Node2Vec [32] focus on learning node embeddings. Similar to Word2Vec [33, 34], they optimize the embedding of a node to predict nodes in a neighborhood. Recursive neural networks [35, 27] and Tree LSTMs [36] consider embeddings of trees, a special type of graphs. Misra & Artzi [37] embed tree representations of typed lambda calculus expressions into vectors, with variable nodes labeled with only their types. This leads to invariance to variable renaming, but is not entirely lossless in terms of semantics. If a formula contains multiple variables of the same type but with different names, it is not possible to know which lambda abstraction binds which variable.

Neural networks on general graphs were first introduced by Gori et al [38] and Scarselli et al [39]. Many follow-up works [40, 41, 42, 43, 44, 45] proposed specific architectures to handle graph-based input by extending recurrent neural network to graph data [38, 41, 42] or making use of graph convolutions based on spectral graph theories [40, 43, 44, 45, 46]. Our approach is most similar to the work of [40], where they encode molecular fragments as neural fingerprints with graph-based convolutions for chemical applications. But to the best of our knowledge, no previous deep learning approaches on general graphs preserve the order of edges. In contrast, we propose a novel way of graph embedding that can preserve the information of edge ordering, and demonstrate its effectiveness for premise selection.

## 3 FormulaNet: Formulas to Graphs to Embeddings

### 3.1 Formulas to Graphs

We consider formulas in higher-order logic [47]. A higher-order formula can be defined recursively based on a vocabulary of constants, variables, and quantifiers. A variable or a constant can act as a value or a function. For example, $\forall f \exists x (f(x, c) \land P(f))$ is a higher-order formula where $\forall$ and $\exists$ are quantifiers, $c$ is a constant value, $P, \land$ are constant functions, $x$ is a variable value, and $f$ is both a variable function and a variable value.

To construct a graph from a formula, we first parse the formula into a tree, where each internal node represents a constant function, a variable function, or a quantifier, and each leaf node represents a variable value or a constant value. We then add edges that connect a quantifier node to all instances of its quantified variables, after which we merge (leaf) nodes that represent the same constant or variable. Finally, for each occurrence of a variable, we replace its original name with VAR, or VARFUNC if it acts as a function. Fig. 2 illustrates these steps.

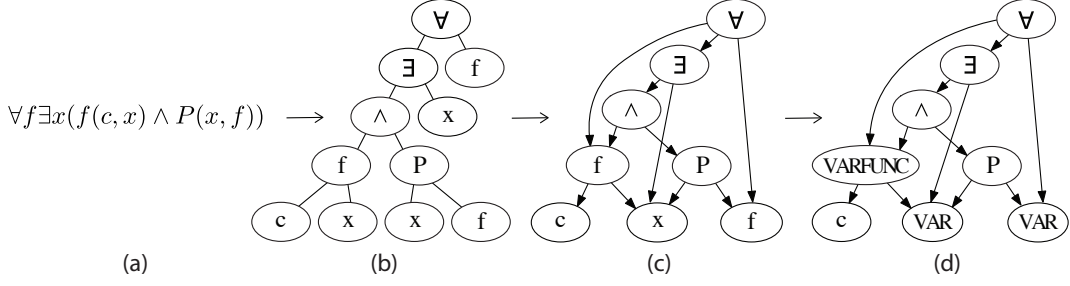

Figure 2: From a formula to a graph: (a) the input formula; (b) parsing the formula into a tree; (c) merging leaves and connecting quantifiers to variables; (d) renaming variables.

Formally, let $\mathcal{S}$ be the set of all formulas, $\mathcal{C}_v$ be the set of constant values, $\mathcal{C}_f$ the set of constant functions, $\mathcal{V}_v$ the set of variable values, $\mathcal{V}_f$ the set of variable functions, and $\mathcal{Q}$ the set of quantifiers. Let $s$ be a higher-order logic formula with no free variables—any free variables can be bounded by adding quantifiers $\forall$ to the front of the formula. The graph $G_s = (V_s, E_s)$ of formula $s$ can be recursively constructed as follows:

- if $s = \alpha$, where $\alpha \in \mathcal{C}_v \cup \mathcal{V}_v$, then $G_s \leftarrow (\{\alpha\}, \emptyset)$, i.e. the graph contains a single node $\alpha$.
- if $s = f(s_1, s_2, \ldots, s_n)$, where $f \in \mathcal{C}_f \cup \mathcal{V}_f$ and $s_1, \ldots, s_n \in \mathcal{S}$, then we perform $G'_s \leftarrow (\bigcup_i^n V_{s_i} \cup \{f\}, \bigcup_i^n E_{s_i} \cup \{(f, \nu(s_i))\}_i)$ followed by $G_s \leftarrow \texttt{MERGE\_C}(G'_s)$, where $\nu(s_i)$ is the "head node" of $s_i$ and $\texttt{MERGE\_C}$ is an operation that merges the same constant (leaf) nodes in the graph.
- if $s = \phi_x t$, where $\phi \in \mathcal{Q}$, $t \in \mathcal{S}$, $x \in \mathcal{V}_v \cup \mathcal{V}_f$, then we perform $G''_s \leftarrow \left(V_t \cup \{f\}, E_t \cup \{(\phi, \nu(t)) \bigcup_{v \in V_t[x]} \{(\phi, v)\}\right)$, followed by $G'_s \leftarrow \texttt{MERGE}_x(G''_s)$ if $x \in \mathcal{V}_v \cup \mathcal{V}_f$ and $G_s \leftarrow \texttt{RENAME}_x(G'_s)$, where $V_t[x]$ is the nodes that represent the variable $x$ in the graph of $t$, $\texttt{MERGE}_x$ is an operation that merges all nodes representing the variable $x$ into a single node, and $\texttt{RENAME}_x$ is an operation that renames $x$ to $\texttt{VAR}$ (or $\texttt{VARFUNC}$ if $x$ acts as a function).

By construction, our graph is invariant to variable renaming, yet no syntactic or semantic information is lost. This is because for a variable node (either as a function or value), its original name in the formula is irrelevant in the graph—the graph structure already encodes where it is syntactically and which quantifier binds it.

## 3.2 Graphs to Embeddings

To embed a graph to a vector, we take an approach similar to performing convolution or message passing on graphs [40]. The overall idea is to associate each node with an initial embedding and iteratively update them. As shown in Fig. 3, suppose $v$ and each node around $v$ has an initial embedding. We update the embedding of $v$ by the node embeddings in its neighborhood. After multi-step updates, the embedding of $v$ will contain information from its local strcuture. Then we max-pool the node embeddings across all of nodes in the graph to form an embedding for the graph.

To initialize the embedding for each node, we use the one-hot vector that represents the name of the node. Note that in our graph all variables have the same name $\texttt{VAR}$ (or $\texttt{VARFUNC}$ if the variable acts as a function), so their initial embeddings are the same. All other nodes (constants and quantifiers) each have their names and thus their own one-hot vectors.

We then repeatedly update the embedding of each node using the embeddings of its neighbors. Given a graph $G = (V, E)$, at step $t + 1$ we update the embedding $x_v^{t+1}$ of node $v$ as follows:

$$x_v^{t+1} = F_P^t\left(x_v^t + \frac{1}{d_v}\left[\sum_{(u,v) \in E} F_I^t(x_u^t, x_v^t) + \sum_{(v,u) \in E} F_O^t(x_v^t, x_u^t)\right]\right), \quad (1)$$

where $d_v$ is the degree of node $v$, $F_I^t$ and $F_O^t$ are update functions using incoming edges and outgoing edges, and $F_P^t$ is an update function to conbine the old embeddings with the new update from neighbor

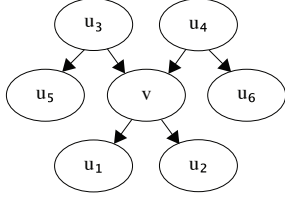

$$x_v^{t+1} = F_P^t\Big(x_v^t + \frac{1}{4}\Big[F_I^t(x_v^t, x_{u_1}^t) + F_I^t(x_v^t, x_{u_2}^t)$$
$$+ F_O^t(x_{u_3}^t, x_v^t) + F_O^t(x_{u_4}^t, x_v^t)\Big]$$
$$+ \frac{1}{3}\Big[F_R^t(x_{u_5}^t, x_{u_3}^t, x_v^t) + F_L^t(x_v^t, x_{u_4}^t, x_{u_6}^t) + F_H^t(x_{u_1}^t, x_v^t, x_{u_2}^t)\Big]\Big)$$

Figure 3: An example of applying the order-preserving updates in Eqn. 2. To update node $v$, we consider its neighbors and its position in all *treelets* (see Sec. 3.3) it belongs to.

nodes. We parametrize these update functions as neural networks; the detailed configurations will be given in Sec. 4.2.

It is worth noting that all node embeddings are updated in parallel using the same update functions, but the update functions can be different across steps to allow more flexibility. Repeated updates allow each embedding to incorporate information from a bigger neighborhood and thus capture more global structures. Interestingly, with zero updates, our model reduces to a bag-of-words representation, that is, a max pooling of individual node embeddings.

To predict the usefulness of a statement for a conjecture, we send the concatenation of their embeddings to a classifier. The classification can also be done in the unconditional setting where only the statement is given; in this case we directly send the embedding of the statement to a classifier. The parameters of the update functions and the classifiers are learned end to end through backpropagation.

### 3.3 Order-Preserving Embeddings

For functions in a formula, the order of its arguments matters. That is, $f(x, y)$ cannot generally be presumed to mean the same as $f(y, x)$. But our current embedding update as defined in Eqn. 1 is invariant to the ordering of arguments. Given that it is possible that the ordering of arguments can be a useful feature for premise selection, we now consider a variant of our basic approach to make our graph embeddings sensitive to the ordering of arguments. In this variant, we update each node considering the ordering of its incoming edges and outgoing edges.

Before we define our new update equation, we need to introduce the notion of a *treelet*. Given a node $v$ in graph $G = (V, E)$, let $(v, w) \in E$ be an outgoing edge of $v$, and let $r_v(w) \in \{1, 2, \ldots\}$ be the rank of edge $(v, w)$ among all outgoing edges of $v$. We define a *treelet* of graph $G = (V, E)$ as a tuple of nodes $(u, v, w) \in V \times V \times V$ such that (1) both $(v, u)$ and $(v, w)$ are edges in the graph and (2) $(v, u)$ is ranked before $(v, w)$ among all outgoing edges of $v$. In other words, a treelet is a subgraph that consists of a head node $v$, a left child $u$ and a right child $w$. We use $\mathcal{T}_G$ to denote all treelets of graph $G$, that is, $\mathcal{T}_G = \{(u, v, w) : (v, u) \in E, (v, w) \in E, r_v(u) < r_v(w)\}$.

Now, when we update a node embedding, we consider not only its direct neighbors, but also its roles in all the treelets it belongs to:

$$x_v^{t+1} = F_P^t\Big(x_v^t + \frac{1}{d_v}\Big[\sum_{(u,v) \in E} F_I^t(x_u^t, x_v^t) + \sum_{(v,u) \in E} F_O^t(x_v^t, x_u^t)\Big]$$
$$+ \frac{1}{e_v}\Big[\sum_{(v,u,w) \in \mathcal{T}_G} F_L^t(x_v^t, x_u^t, x_w^t) + \sum_{(u,v,w) \in \mathcal{T}_G} F_H^t(x_u^t, x_v^t, x_w^t) + \sum_{(u,w,v) \in \mathcal{T}_G} F_R^t(x_u^t, x_w^t, x_v^t)\Big]\Big)$$
$$(2)$$

where $e_v = |\{(u, v, w) : (u, v, w) \in \mathcal{T}_G \vee (v, u, w) \in \mathcal{T}_G \vee (u, w, v) \in \mathcal{T}_G\}|$ is the number of total treelets containing $v$. In this new update equation, $F_L$ is an update function that considers a treelet where node $v$ is the left child. Similarly, $F_H$ considers a treelet where node $v$ is the head and $F_R$ considers a treelet where node $v$ is the right child. As in Sec. 3.2, the same update functions are applied to all nodes at each step, but across steps the update functions can be different. Fig. 3 shows the update equation of a concrete example.

Our design of Eqn. 2 now allows a node to be embedded differently dependent on the ordering of its own arguments and dependent on which argument slot it takes in a parent function. For example, the function node $f$ can now be embedded differently for $f(a, b)$ and $f(b, a)$ because of the output of $F_H$ can be different. As another example, in the formula $g(f(a), f(a))$, there are two function

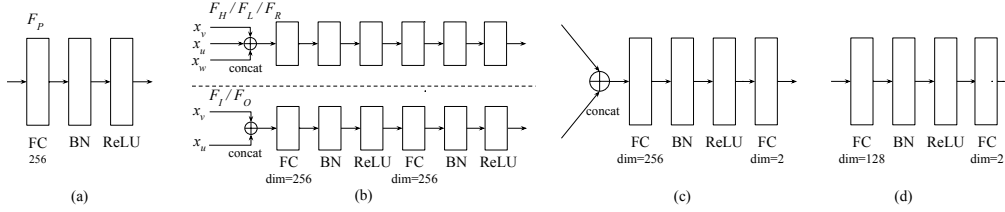

Figure 4: Configurations of the update functions and classifiers: (a) $F_P$ in Eqn. 1 and 2; (b) $F_I, F_O$ in Eqn. 1 and 2, and $F_L, F_H, F_R$ in Eqn. 2; (c) conditional classifier; (d) unconditional classifier.

nodes with the same name $f$, same parent $g$, and same child $a$, but they can be embedded differently because only $F_L$ will be applied to the $f$ as the first argument of $g$ and only $F_R$ will be applied to the $f$ as the second argument of $g$.

To distinguish the two variants of our approach, we call the method with the treelet update terms *FormulaNet*, as opposed to the basic *FormulaNet-basic* without considering edge ordering.

## 4    Experiments

### 4.1    Dataset and Evaluation

We evaluate our approach on the HolStep dataset [10], a recently introduced benchmark for evaluating machine learning approaches for theorem proving. It was constructed from the proof trace files of the HOL Light theorem prover [7] on its multivariate analysis library [48] and the formal proof of the Kepler conjecture. The dataset contains 11,410 conjectures, including 9,999 in the training set and 1,411 in the test set. Each conjecture is associated with a set of statements, each with a ground truth label on whether the statement is useful for proving the conjecture. There are 2,209,076 conjecture-statement pairs in total. We hold out 700 conjectures from the training set as the validation set to tune hyperparameters and perform ablation analysis.

Following the evaluation setup proposed in [10], we treat premise selection as a binary classification task and evaluate classification accuracy. Also following [10], we evaluate two settings, the *conditional* setting where both the conjecture and the statement are given, and the *unconditional* setting where the conjecture is ignored. In HolStep, each conjecture is associated with an equal number of positive statements and negative statements, so the accuracy of random prediction is $50\%$.

### 4.2    Network Configurations

The initial one-hot vector for each node has 1909 dimensions, representing 1909 unique tokens. These 1909 tokens include 1906 unique constants from the training set and three special tokens, "VAR", "VARFUNC", and "UNKNOWN" (representing all novel tokens during testing). We use a linear layer to map one-hot encodings to 256-dimensional vectors. All of the following intermediate embeddings are 256-dimensional.

The update functions in Eqn. 1 and Eqn. 2 are parametrized as neural networks. Fig. 4 (a), (b) shows their configurations. All update functions are configured the same: concatenation of inputs followed by two fully connected layers with ReLUs, Batch Normalizations [49].

The classifier for the conditional setting takes in the embeddings from the conjecture and the statement. Its configuration is shown in Fig. 4 (c). The classifier for the unconditional setting uses only the embedding of the statement; its configuration is shown in Fig. 4 (d).

### 4.3    Model Training

We train our networks using RMSProp [50] with 0.001 learning rate and $1 \times 10^{-4}$ weight decay. We lower the learning rate by 3X after each epoch. We train all models for five epochs and all networks converge after about three or four epochs.

It is worth noting that there are two levels of batching in our approach: intra-graph batching and inter-graph batching. Intra-graph batching arises from the fact that to embed a graph, each update

Table 1: Classification accuracy on the test set of our approach versus baseline methods on HolStep in the unconditional setting (conjecture unknown) and the conditional setting (conjecture given).

|  | CNN [10] | CNN-LSTM [10] | FormulaNet-basic | FormulaNet |
|---|---|---|---|---|
| Unconditional | 83 | 83 | 89.0 | **90.0** |
| Conditional | 82 | 83 | 89.1 | **90.3** |

function ($F_P, F_I, F_O, F_L, F_H, F_R$ in Eqn. 2) is applied to all nodes in parallel. This is the same as training each update function as a standalone network with a batch of input examples. Thus regular batch normalization can be directly applied to the inputs of each update function *within a single graph*, as shown in Fig. 4(a)(b).

Furthermore, this batch normalization within a graph can be run in the training mode even when we are only performing inference to embed a graph, because there are multiple input examples to each update function within a graph. Another level of batching is the regular batching of multiple graphs in training, as is necessary for training the classifier. As usual, batch normalization across graphs is done in the evaluation mode in test time.

We also apply intermediate supervision after each step of embedding update using a separate classifier. For training, our loss function is the sum of cross-entropy losses for each step. We use the prediction from the last step as our final predictions.

## 4.4 Main Results

Table 1 compares the accuracy of our approach versus the best existing results [10]. Our approach improves the best existing result by a large margin from 83% to 90.3% in the conditional setting and from 83% to 90.0% in the unconditional setting. We also see that FormulaNet gives a 1% improvement over the FormulaNet-basic, validating our hypothesis that the order of function arguments provides useful cues.

Consistent with prior work [10], conditional and unconditional selection have similar performances. This is likely due to the data distribution in HolStep. In the training set, only 0.8% of the statements appear in both a positive statement-conjecture pair and a negative statement-conjecture pair, and the upper performance bound of unconditional selection is 97%. In addition, HolStep contains 9,999 unique conjectures but 1,304,888 unique statements for training, so it is likely easier for the network to learn useful patterns from statements than from conjectures.

We also apply Deepwalk [30], an unsupervised approach for generating node embeddings that is purely based on graph topology without considering the token associated with each node. For each formula graph, we max-pool its node embeddings and train a classifier. The accuracy is 61.8% (conditional) and 61.7% (unconditional). This result suggests that for embedding formulas it is important to use token information and end-to-end supervision.

## 4.5 Ablation Experiments

**Invariance to Variable Renaming** One motivation for our graph representation is that the meaning of formulas should be invariant to the renaming of variable values and variable functions. To achieve such invariance, we perform two main transformations of a parse tree to generate a graph: (1) we convert the tree to a graph by linking quantifiers and variables, and (2) we discard the variable names.

We now study the effect of these steps on the premise selection task. We compare FormulaNet-basic with the following three variants whose only difference is the format of the input graph:

- *Tree-old-names*: Use the parse tree as the graph and keep all original names for the nodes. An example is the tree in Fig. 2 (b).

- *Tree-renamed*: Use the parse tree as the graph but rename all variable values to `VAR` and variable functions to `VARFUNC`.

- *Graph-old-names*: Use the same graph as FormulaNet-basic but keep all original names for the nodes, thus making the graph embedding dependent on the original variable names. An example is the graph in Fig. 2 (c).

Table 2: The accuracy of FormulaNet-basic and its ablated versions on original and renamed validation set.

|  | Tree-old-names | Tree-renamed | Graph-old-names | Our Graph |
|---|---|---|---|---|
| Original Validation | 89.7 | 84.7 | 89.8 | 89.9 |
| Renamed Validation | 82.3 | 84.7 | 83.5 | 89.9 |

Table 3: Validation accuracy of proposed models with different numbers of update steps on conditional premise selection.

| Number of steps | 0 | 1 | 2 | 3 | 4 |
|---|---|---|---|---|---|
| FormulaNet-basic | 81.5 | 89.3 | 89.8 | 89.9 | 90.0 |
| FormulaNet | 81.5 | 90.4 | 91.0 | 91.1 | 90.8 |

We train these variants on the same training set as FormulaNet-basic. To compare with FormulaNet-basic, we evaluate them on the same held-out validation set. In addition, we generate a new validation set (Renamed Validation) by randomly permutating the variable names in the formulas—the textual representation is different but the semantics remains the same. We also compare all models on this renamed validation set to evaluate their robustness to variable renaming.

Table 2 reports the results. If we use a tree with the original names, there is a slight drop when evaluate on the original validation set, but there is a very large drop when evaluated on the renamed validation set. This shows that there are features exploitable in the original variable names and the model is exploiting it, but the model is essentially overfitting to the bias in the original names and cannot generalize to renamed formulas. The same applies to the model trained on graphs with the original names, whose performance also drops drastically on renamed formulas.

It is also interesting to note that the model trained on renamed trees performs poorly, although it is invariant to variable renaming. This shows that the syntactic and semantic information encoded in the graph on variables—particularly their quantifiers and coreferences—is important.

## 4.6 Visualization of Embeddings

**Number of Update Steps** An important hyperparameter of our approach is the number of steps to update the embeddings. Zero steps can only embed a bag of unstructured tokens, while more steps can embed information from larger graph structures. Table 3 compares the accuracy of models with different numbers of update steps. Perhaps surprisingly, models with zero steps can already

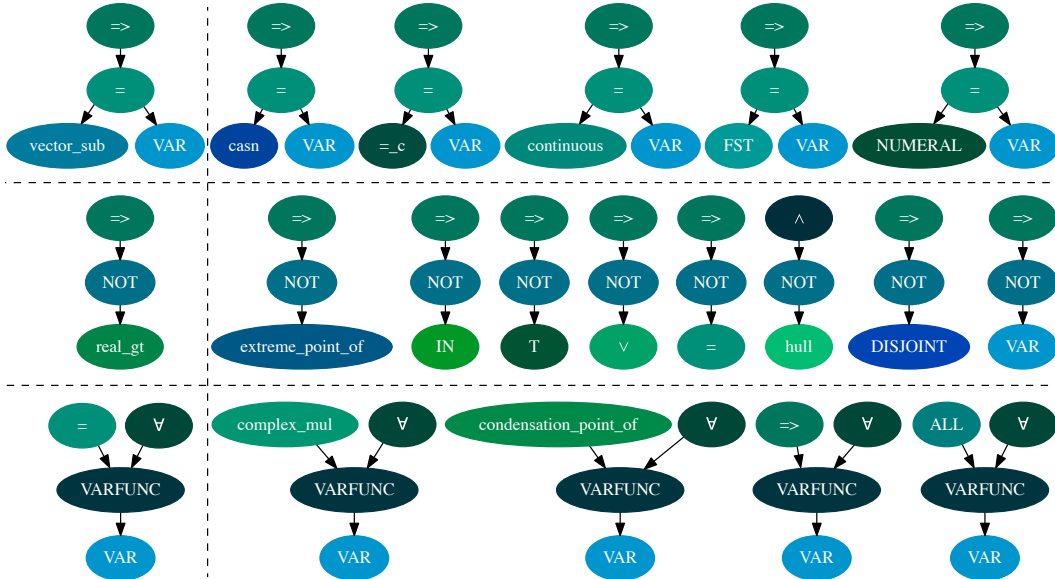

Figure 5: Nearest neighbors of node embeddings after step 1 with FormulaNet. Query nodes are in the first column. The color of each node is coded by a t-SNE [51] projection of its step-0 embedding into 2D. The closer the colors, the nearer two nodes are in the step-0 embedding space.

achieve an accuracy of $81.5\%$, showing that much of the performance comes from just the names of constant functions and values. More steps lead to notable increases of accuracy, showing that the structures in the graph are important. There is a diminishing return after 3 steps, but this can be reasonably expected because a radius of 3 in a graph is a fairly sizable neighborhood and can encompass reasonably complex expressions—a node can influence its grand-grandchildren and grand-grandparents. In addition, it would naturally be more difficult to learn generalizable features from long-range patterns because they are more varied and each of them occurs much less frequently.

To qualitatively examine the learned embeddings, we find out a set of nodes with similar embeddings and visualize their local structures in Fig. 5. In each row, we use a node as the query and find the nearest neighbors across all nodes from different graphs. We can see that the nearest neighbors have similar structures in terms of topology and naming. This demonstrates that our graph embeddings can capture syntactic and semantic structures of a formula.

## 5 Conclusion

In this work, we have proposed a deep learning-based approach to premise selection. We represent a higher-order logic formula as a graph that is invariant to variable renaming but fully preserves syntactic and semantic information. We then embed the graph into a continuous vector through a novel embedding method that preserves the information of edge ordering. Our approach has achieved state-of-the-art results on the HolStep dataset, improving the classification accuracy from 83% to 90.3%.

**Acknowledgements**  This work is partially supported by the National Science Foundation under Grant No. 1633157.

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
