[Reviews · NeurIPS 2017]

Reviewer 1



This paper applies deep learning to the problem of premise selection in HOL theorem proving. The paper is clear (with some issues listed below) and sophisticated in deep learning methods with an appropriate experimental methodology and thorough discussion of related work. However, the main weakness in my opinion is the results. While an improvement from 83% percent accuracy to 91% is reported, the results still exhibit the bizarre property that the performance is the same whether or not one uses the information of what the system is trying to prove. There is no discussion of this bizarre aspect of the experiments. Is the system only able to identify domain-independent lemmas that are always useful --- like basic properties of sets and functions? I also have some technical questions. The symbols (words) are initialized with one-hot vectors. Word embeddings for natural language applications typically have hundreds of dimensions. It would be nice to know how many distinct constants the corpus contains and hence what is the dimension of the embedding vectors being used. Also I found the discussion of intra-graph batching interesting but would like some additional information. It seems that intra-example batching would apply to vision CNNs as well as each filter of each layer is repeatedly applied across a single convolution. My expectation is that this is not typically done because it is not supported by frameworks such as tensor flow where BN is a layer and minibatching is standardized to be be inter-instance. Did the authors implement novel layers or framework features? Finally, at an intuitive level one would expect relevance to be driven by types. To prove a graph theory theorem one should use graph theory lemmas where "graph" is a datatype. Was there any attempt to take this intuition into account?

Reviewer 2



The paper addresses the problem of premise selection for theorem proving. The objective in this line of work is to learn a vector representation for the formulas. This vector representation is then used as the input to a downstream machine learning model performing premise selection. The contributions of the paper are: (1) a representation of formulas as graph; (2) learning vector representations for the nodes and ultimately for the formula graphs; and (3) show empirically that this representation is superior to the state of the art on theorem proving data sets. Pros * The paper is very well written. A pleasure to read and despite its technical content highly accessible. * The experimental results show a significant improvement over the state of the art. * The authors discuss all related work I'm aware of and do it in a fair and comprehensive way. Cons * There has been previous work on this topic. The novelty here is the graph representation and the way in which node embeddings are learned. The former is a nice idea. The latter is more interesting from a ML point of view but it would have made the paper much stronger if the authors had compared the averaging of neighboring node embeddings approach to other node embedding approaches such as DeepWalk. This has been added after the rebuttal: The main weakness of the paper is in my opinion the (lack of) novelty of the proposed approach. The difference to previous work is the graph representation of the formulas. To represent formulas as graphs is not new. Now, instead of learning a vector representation for the formulas directly, a vector representation of the graphs is learned. The downstream architectures are also more or less what has been done in previous work. It's a nice idea for premise selection that performs better than the SOTA by combining existing ideas (graph representation of formulas + graph vector representation learning). Since there's nothing really new here I would have liked more experimental results and a comparison of different graph embedding methods (many of these are cited by the authors). Minor comments: Line 154: "To initialize the embedding for each node, we use the one-hot vector that represents the name of the node." You probably mean that you initialize the embeddings randomly?

Reviewer 3



This paper proposes a deep learning approach to learn representation of higher-order logic formulae in the context of theorem proving. Formulae are first converted into directed graphs, from which the embeddings are derived. The technique is reminiscent of recursive neural networks (RNNs) but to the best of my knowledge RNNs have never be applied to higher order logical formulae before. The paper is well written and the proposed approach well motivated. Experiments show the effectiveness of this deep embedding technique. I have some minor comments: 1. The paper that first introduced RNNs had the goal of learning representations of logical terms. It is so closely related to this work that it should be cited: C. Goller and A. K{\"u}chler. Learning task-dependent distributed structure-representations by backpropagation through structure. In IEEE International Conference on Neural Networks, pages 347--352, 1996. 2. It seems to me that the formula->graph transformation always yield acyclic graphs but this should be perhaps said explicitly (or give counterexamples). 3. I have problems with the example at the beginning of 3.1. Functions normally return objects, not predicates, therefore f(x,c) /\ P(f) does not seem to be well formed: what is the conjunction of a Boolean value --- P(f) --- and an object?? Perhaps it should be something like \forall f \exists x Q(f(x,c)) /\ P(f) 4. You talk about higher-order logic but it seems you are just using a subset of second-order logic. Can you quantify over predicates? A brief synopsis of the syntax and semantics of the logic assumed in the paper would be helpful. 5. Second-order logic is undecidable, is there any associated impact on theory behind your work? Related to this, Siegelman & Sontag showed that recurrent networks with arbitrary precision weights have super-Turing computational power, are there any links between these results and your work?